# Urinary 5-HIAA as a Potential Diagnostic Marker in Acute Appendicitis: A Preliminary Report of Its Promising Role in Early Detection

**DOI:** 10.3390/medicina61061070

**Published:** 2025-06-11

**Authors:** Murat Demir, Alper Gumus, Huseyin Kilavuz, Feyyaz Gungor, Sibel Yaman, Baki Ekci, Idris Kurtulus

**Affiliations:** 1Department of General Surgery, Basaksehir Cam and Sakura City Hospital, University of Health Sciences, Istanbul 34480, Turkey; drhuseyinkilavuz@gmail.com (H.K.); feyyaz.gngr@gmail.com (F.G.); syaman.ctf@hotmail.com (S.Y.); drbaki@yahoo.com (B.E.); idriskurtulus@gmail.com (I.K.); 2Department of Biochemistry, Basaksehir Cam and Sakura City Hospital, University of Health Sciences, Istanbul 34480, Turkey; dralpergumus@gmail.com

**Keywords:** acute appendicitis, 5-hydroxyindoleacetic acid, alvarado score, diagnostic accuracy

## Abstract

***Background***: Acute appendicitis (AA) is a common surgical emergency worldwide. Over the past few decades, diagnostic imaging has become a cornerstone in the identification of acute appendicitis, significantly contributing to the reduction in unnecessary laparotomies and associated healthcare costs. This study aimed to investigate the influence of serum and spot urine 5-hydroxyindoleacetic acid (5-HIAA) levels, as well as other established clinical and biochemical parameters on the diagnosis of acute appendicitis. ***Methods***: This prospective study was conducted between January and November 2023, evaluating 97 patients diagnosed with acute appendicitis. Serum and spot urine 5-HIAA levels, level of white blood cell (WBC), neutrophils, lymphocytes, platelets, C-reactive protein (CRP), and Alvarado score were analyzed. Patients were further allocated to subgroups based on their Alvarado scores, the onset time of the symptoms, and pathological findings to statistically assess the relationship between the parameters. ***Results***: The mean age of the patients was 34.6 ± 14.8 years. Of the patients, 57 (58.8%) were male, and 40 (41.2%) were female. Spot urine 5-HIAA levels exhibited statistically significant variation among different symptom onset time groups, with elevated levels observed in patients presenting within the first 12 h of symptom onset (*p* < 0.001). Neutrophil counts were significantly different among Alvarado score groups (*p* < 0.001), whereas CRP levels significantly increased with the onset time of the symptoms (*p* < 0.001). ***Conclusions***: Increased spot urine 5-HIAA is supportive of the diagnosis of AA in patients presenting within the first 12 h of symptom onset. Hematological parameters, especially CRP, may provide more reliable information regarding disease severity and progression.

## 1. Introduction

Acute appendicitis (AA) is one of the most common surgical emergencies and its annual incidence ranges from 96.5 to 100 per 100,000 adults [1]. Acute appendicitis can lead to complications like perforation, peritonitis, and sepsis. Early diagnosis and surgical intervention are important to reduce complication risks. Diagnosis is primarily based on clinical findings, combined with imaging techniques and laboratory analyses [2,3]. The Alvarado score is a useful tool for the early diagnosis of AA. It assesses various signs, symptoms, and laboratory findings based on their specificity, sensitivity, predictive value, and probability. This scoring system considers eight important factors in diagnosing acute appendicitis: localized tenderness in the right lower quadrant, leukocytosis, migration of pain, shift to the left, temperature elevation, nausea/vomiting, anorexia-acetone, and direct rebound pain. Using these factors, the Alvarado score helps to interpret the complex clinical presentation of acute appendicitis [4,5].

Serotonin is a neurotransmitter that plays an important role in the human body. While it is mainly produced in the brain, about 90% of serotonin is found in the gastrointestinal tract, where it helps control intestinal movements. Enterochromaffin cells, which store serotonin, are mainly found in the appendix. The main breakdown product of serotonin is 5-Hydroxyindoleacetic acid (5-HIAA). When there is inflammation, these cells release serotonin, which is then turned into 5-HIAA [6,7,8]. This process is thought to be important in diagnosing various inflammatory and cancer-related conditions, such as acute appendicitis.

Diagnostic tools like the Alvarado score and imaging methods help the diagnosis of acute appendicitis. However, sometimes additional examinations or tests are needed. These extra tests help achieve an accurate diagnosis. This study aimed to investigate the influence of serum and spot urine 5-HIAA levels, as well as other established clinical and biochemical parameters, on the diagnosis of acute appendicitis.

## 2. Materials and Methods

### 2.1. Patients and Samples

The ethical approval for this study was obtained from the Ethics Committee of Cam and Sakura City Hospital (Approval Code: 2022/228; Approval Date: 7 July 2022). The study was conducted in accordance with the principles outlined in the Helsinki Declaration. Written informed consent was obtained from all participants, comprising 97 patients diagnosed with acute appendicitis who voluntarily participated in the study. Patients were divided into subgroups based on their Alvarado scores (Group 1: Scores 1–4; Group 2: Scores 5–6; Group 3: Scores 7–9), the onset time of the symptoms (Group 1: <12 h; Group 2: 12–24 h; Group 3: 24–48 h; Group 4: >48 h), and histopathological findings (Group 1: gangrenous appendicitis; Group 2: non-gangrenous appendicitis).

Patients over the age of 18 who were clinically and radiologically diagnosed with acute appendicitis were included in the study. Patients who were using medications that affect serotonin levels (such as serotonin and norepinephrine reuptake inhibitors, antidepressants, monoamine oxidase inhibitors, and lithium), pregnant patients, patients with coagulopathy, immunosuppressed patients, patients with acute gastroenteritis symptoms (diarrhea), and those with incomplete data were excluded from the study.

Demographic characteristics, including age and gender, as well as clinical findings such as symptom onset time, right lower quadrant pain, and rebound tenderness, were recorded for each patient. Blood and urine samples were collected during initial admission, prior to surgery to evaluate serum 5-HIAA, urinary 5-HIAA, complete blood count (CBC), C-reactive protein (CRP), and serum creatinine levels.

Serum samples were obtained by collecting blood into gel separator tubes and centrifuging at 1500 cycles/min for 15 min. Serum and urine samples for 5-HIAA analysis were stored at −80 °C until the day of analysis. For CBC analysis, blood was collected into K_2_EDTA-containing tubes, and no pre-processing was performed on these whole blood samples. CRP and CBC measurements were conducted directly on the collected samples. Additionally, tissue samples obtained during surgery were evaluated in the pathology laboratory.

### 2.2. Laboratory Methods

Urinary 5-HIAA level was determined chromatographically using the LC-MS/MS method with a Shimadzu LCMS-8045 analyzer (Shimadzu Corporation, Kyoto, Japan) (Tokyo, Japan) from spot urine samples. The results were reported as ratios normalized to urinary creatinine levels to ensure comparability among patients and were reported in units of mg/g creatinine. Serum 5-HIAA level was analyzed using an ELISA method with a Cloud Clone Group test kit (Cloud-Clone Corporation, Wuhan, China). The intra-assay coefficient of variation (CV) was <10%, and the inter-assay CV was <12%. The results were reported in units of ng/mL.

The concentration of CRP levels was measured by a spectrophotometrical method using a Roche Cobas c702 clinical chemistry analyzer (Roche Diagnostics, Rotkreuz, Switzerland).

CBC was measured with an electronic automated cell counter using a Sysmex XN-1000 automatic hematology analyzer (Sysmex Corporation, Kobe, Japan).

The final diagnosis in all operated cases was confirmed by histological examination. The histopathology results of appendectomy specimens were categorized as gangrenous appendicitis (GA) and non-gangrenous appendicitis (NGA).

### 2.3. Statistical Analysis

The power analysis using G*Power 3.1 software (Heinrich Heine University, Düsseldorf, Germany) revealed an effect size (*f*) of 0.33, which corresponds to a medium-to-large effect. With a total sample size of 97 participants distributed across three groups, the statistical power of the study was calculated to be approximately 83.3%. This indicates that the study has adequate power to detect significant differences among the groups, minimizing the risk of a Type II error.

The statistical analysis was performed using SPSS version 21 (IBM, New York, NY, USA). Descriptive statistics, including minimum, maximum, mean, and standard deviation, were calculated. The normality of data distribution was assessed by examining histograms, kurtosis, and skewness values. The skewness and kurtosis values for all parameters were observed to be below 2, indicating that the data conforms to the characteristics of a normal distribution. Parameters demonstrating normal distribution characteristics were analyzed using Student’s *t*-test or ANOVA, depending on the number of subgroups, with statistical significance set at *p* < 0.05. Relationships between parameters were evaluated using Pearson correlation analysis. The diagnostic performance of parameters was assessed using receiver operating characteristic (ROC) curve analysis.

## 3. Results

A total of 97 patients were included in the study. The mean age of the patients was 34.6 ± 14.8 years. Of the participants, 57 (58.8%) were male and 40 (41.2%) were female.

The comparison of the results among the Alvarado groups indicated that there were no significant differences in the evaluated biochemical parameters, except for the neutrophil count (Table 1) (*p* > 0.05). Neutrophil counts were significantly different among Alvarado score groups (*p* = 0.027). The lowest neutrophil count was found in Group 3 (Alvarado 7–9) (13.20 ± 1.78 × 10^3^/μL).

The comparison of the results among the symptom onset time groups revealed that there was a statistically significant difference among the spot urine 5-HIAA levels and CRP values of the groups (*p* < 0.05). No statistically significant difference was observed among the other biochemical parameters (*p* > 0.05) (Table 2). Accordingly, the highest urinary 5-HIAA level was detected in Group 1 (onset less than 12 h) (5.66 ± 3.13 mg/g crea), whereas the lowest value was in Group 4 (onset more than 48 h) (3.23 ± 2.39 mg/g crea) (*p*: 0.022). CRP levels increased progressively starting with the shortest symptom onset time, from 25.00 ± 40.67 mg/L in Group 1 to the longest, 94.72 ± 117.84 mg/L, found in Group 4. This increase was found statistically significant (*p* = 0.004) (Table 2).

The comparison of the results among the histopathology groups revealed that there were no significant differences in the evaluated biochemical parameters, except for the CRP level (Table 3) (*p* > 0.05). CRP levels were significantly higher in the gangrenous appendicitis group (141.90 ± 147.86 mg/L) compared to the non-gangrenous appendicitis group (41.49 ± 53.67 mg/L) (*p* < 0.001).

## 4. Discussion

Acute appendicitis refers to the sudden and severe inflammation of the appendix. It is the leading cause of abdominal surgery with patients typically presenting acutely within the first 24 h from the onset of symptoms. Appendectomy is the most common abdominal surgical emergency worldwide. Early diagnosis and timely surgical intervention are crucial to minimize the risk of complications [9]. The diagnosis is primarily based on clinical findings, supported by imaging techniques and laboratory analyses [2,3]. Furthermore, the Alvarado score is a valuable tool for the early diagnosis of acute appendicitis. This score evaluates various signs, symptoms, and laboratory findings in terms of specificity, sensitivity, predictive value, and probability. It includes eight key factors in the diagnosis of acute appendicitis: localized tenderness in the right lower quadrant, leukocytosis, migration of pain, shift to the left, temperature elevation, nausea and vomiting, anorexia and acetone, and direct rebound tenderness. By integrating these factors, the Alvarado score assists in interpreting the complex clinical presentation of acute appendicitis [4,5]. The maximum possible score for suspected acute appendicitis is 9 points. An Alvarado score of 7 or higher is significantly associated with the presence of acute appendicitis [10]. However, if appendicitis is left untreated or diagnosed of late, patients face a risk of appendiceal perforation, abscess formation, peritonitis, sepsis, and death. In some cases, additional tests or examinations are necessary to achieve an accurate diagnosis. Therefore, this study aimed to investigate whether serum or urine 5-HIAA, a serotonin metabolite found in the gastrointestinal tract and known to increase during inflammation, could assist in the diagnosis of acute appendicitis.

Serotonin is mainly secreted by enterochromaffin cells in the gastrointestinal system. The appendix is rich in enterochromaffin cells [7,8]. 5-Hydroxy indoleacetic acid (5-HIAA) is the primary metabolite of serotonin and is excreted with urine. For this reason, 5-HIAA levels have been investigated as an aid in the diagnosis of acute appendicitis, but a consensus has not yet been reached. This study investigated the potential role of serum and urine 5-HIAA levels in the diagnosis of acute appendicitis. Jangjoo et al. showed an approximately 44% sensitivity and 81% specificity regarding the validity of urinary 5-HIAA level for the diagnosis for acute appendicitis [11]. Sample preservation is an important point to consider when assessing urinary 5-HIAA levels. It is commonly recommended that urine 5-HIAA should be preserved in acidified urine and kept away from light [12]. Bolandparvaz et al. [13] measured urinary 5-HIAA level using high performance liquid chromatography (HPLC) and concluded that high urinary 5-HIAA level is related with acute appendicitis. Similarly, the present study’s results revealed that urine 5-HIAA levels were elevated in patients who presented within the first 12 h. Furthermore, urine 5-HIAA levels remained elevated for the first 12 h following symptom onset, gradually decreasing thereafter. So, the results showed a statistically significant difference in urinary 5-HIAA levels among patients with different symptom onset times, particularly demonstrating the highest levels in the <12 h group. This trend suggests that serotonin is released in greater quantities from enterochromaffin cells in the appendix during the initial phase of inflammation, whereas its release decreases in the later stages due to factors such as tissue edema, venous obstruction, or necrosis.

Some studies in the literature have reported higher urinary 5-HIAA levels in perforated appendicitis cases compared to non-perforated ones [14]. However, findings from many prospective studies and experimental animal models are consistent with this study’s results, showing that spot urinary 5-HIAA levels are high in the early stages of acute appendicitis but decrease as the disease progresses and necrosis develops [15,16,17]. Moreover, the present study showed that as the inflammation progressed to gangrene of the appendix, the urinary concentration of 5-HIAA decreased. This suggests that elevated CRP levels may be associated with the presence of gangrenous appendicitis. Prior studies investigating the association between symptom onset time and urinary 5-HIAA levels also supported this hypothesis, with higher levels detected in the early phase of appendicitis and decreasing levels in later stages [15]. Conversely, studies that did not consider symptom onset time have often failed to demonstrate significant findings regarding urinary 5-HIAA levels, likely due to this variability [18,19]. According to the present study, this decrease could be a warning sign of perforation of the appendix.

Many studies in the literature discuss whether the urinary 5-HIAA is a reliable parameter in the diagnosis of acute appendicitis [11,18,19,20,21]. A study by Adaway et al. [22] demonstrated that serum 5-HIAA levels remain unchanged when the estimated glomerular filtration rate (eGFR) is above 60 mL/min/1.73 m^2^ but increase once eGFR falls below this threshold. Despite variations in spot urinary 5-HIAA levels during the inflammatory process, no significant changes in serum 5-HIAA levels were found in the present study. The possible explanation of this situation might be the increased renal clearance of 5-HIAA following its release from the appendix. The lack of correlation between serum and urinary 5-HIAA levels supports this physiological explanation.

CRP was discovered by Tillett and Francis in 1930. CRP, a protein produced by the liver, is released in response to inflammation in the body. Significantly elevated CRP levels are typically associated with infectious causes. High CRP levels in patients with symptoms and signs of acute appendicitis are also associated with greater inflammation of the appendix. Once the inflammatory response begins, CRP concentration rises rapidly and can increase up to a thousand times within just one day [23,24,25,26]. CRP levels demonstrated significant differences across both symptom duration groups and necrosis-based classifications, reinforcing its role as a strong inflammatory marker [27]. In the present study, high CRP levels were seen in patients diagnosed with acute appendicitis gradually increasing with the symptom onset time. In addition, high CRP levels were also associated with the histopathological status of the appendix, with the highest levels seen in gangrenous appendicitis.

While this study provides valuable insights, it is important to acknowledge certain limitations. Notably, the lack of a control group, the absence of established threshold values for serum and urine 5-HIAA levels in healthy individuals, and the undetermined cutoff values for identifying pathological conditions are considered limitations of the present study.

## 5. Conclusions

In this study, which investigated the role of the serotonin metabolite 5-HIAA in the diagnosis of acute appendicitis, it was found that urinary 5-HIAA levels significantly increased in patients who presented within the first 12 h of symptom onset. However, it is important to note that these data alone are not sufficient as diagnostic markers. Therefore, further studies are required to determine the threshold values of urinary 5-HIAA in healthy individuals and to establish cutoff values for an accurate diagnosis.

## Figures and Tables

**Table 1 medicina-61-01070-t001:** Comparison of biochemical parameters among Alvarado score groups.

Parameters	Group 1 (Alvarado 1–4) (n: 33)	Group 2 (Alvarado 5–6) (n: 36)	Group 3 (Alvarado 7–9) (n: 28)	*p* Value
Urinary 5-HIAA (mg/g crea)	4.50 ± 2.80	4.08 ± 2.54	5.04 ± 3.03	0.39
Serum 5-HIAA (ng/mL)	12.81 ± 4.92	13.13 ± 4.99	13.36 ± 4.61	0.85
WBC (×10^3^/μL)	10.06 ± 4.66	12.16 ± 13.61	10.23 ± 4.65	0.89
Neutrophils (×10^3^/μL)	13.80 ± 1.90	13.83 ± 2.17	13.20 ± 1.78	**0.027**
Lymphocytes (×10^3^/μL)	1.69 ± 0.68	2.34 ± 1.24	1.98 ± 0.89	0.57
Platelets (×10^3^/μL)	237.90 ± 64.60	263.30 ± 57.09	278.10 ± 97.62	0.09
CRP (mg/L)	61.90 ± 93.40	42.09 ± 60.51	52.47 ± 66.79	0.54

**Table 2 medicina-61-01070-t002:** Comparison of biochemical parameters based on symptom onset time.

Parameters	Group 1 (<12 h) (n: 29)	Group 2 (13–24 h) (n: 25)	Group 3 (25–48 h) (n: 24)	Group 4 (>48 h) (n: 19)	*p* Value
Urinary 5-HIAA (mg/g crea)	5.66 ± 3.13	4.42 ± 2.80	4.20 ± 2.08	3.23 ± 2.39	**0.022**
Serum 5-HIAA (ng/mL)	13.65 ± 3.57	12.65 ± 4.12	12.75 ± 5.26	13.23 ± 6.70	0.73
WBC (×10^3^/μL)	10.59 ± 3.58	9.59 ± 4.10	9.86 ± 5.27	14.38 ± 18.49	0.87
Neutrophils (×10^3^/μL)	14.05 ± 1.66	13.72 ± 2.04	13.81 ± 2.41	12.68 ± 1.66	0.85
Lymphocytes (×10^3^/μL)	1.90 ± 0.71	2.11 ± 0.99	1.97 ± 1.00	2.12 ± 1.42	0.30
Platelets (×10^3^/μL)	259.00 ± 65.45	260.68 ± 98.93	252.00 ± 66.23	263.32 ± 64.02	0.96
CRP (mg/L)	25.00 ± 40.67	33.67 ± 50.00	69.20 ± 68.32	94.72 ± 117.84	**0.004**

**Table 3 medicina-61-01070-t003:** Comparison of biochemical parameters based on histopathological status.

Parameters	Group 1 (Without Necrosis) (n: 87)	Group 2 (With Necrosis) (n: 10)	*p* Value
Urinary 5-HIAA (mg/g crea)	4.59 ± 2.78	3.75 ± 2.74	0.47
Serum 5-HIAA (ng/mL)	11.13 ± 5.16	9.90 ± 4.89	0.36
WBC (×10^3^/μL)	12.87 ± 4.72	15.05 ± 5.46	0.17
Neutrophils (×10^3^/μL)	10.75 ± 9.40	12.19 ± 5.21	0.63
Lymphocytes (×10^3^/μL)	2.06 ± 1.03	1.66 ± 0.87	0.24
Platelets (×10^3^/μL)	262.07 ± 76.14	232.10 ± 52.03	0.22
CRP (mg/L)	41.49 ± 53.67	141.90 ± 147.86	**<0.001**

## Data Availability

All data and materials are available and the corresponding author is Demir M.

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
