# Peer review of "Urinary 5-HIAA as a Potential Diagnostic Marker in Acute Appendicitis: A Preliminary Report of Its Promising Role in Early Detection"

_medicina, 2025, doi:10.3390/medicina61061070_

Round 1
Reviewer 1 Report
Comments and Suggestions for Authors
• What is the central question addressed by the research?
The main question addressed refers to the utility of urinary 5HIAA as a diagnostic marker in the early detection of acute appendicitis.
• Do you consider the topic original or relevant to the field?
The first published papers that discuss the relevance of urinary 5-HIAA in acute appendicitis date back about 15 years. Since then, the problem has periodically resurfaced, and the issue of the utility of this test for diagnosis is far from being clarified. So, this paper tries to shed more light on this domain.
• What does it add to the subject area compared with other published material?
A relevant study concerns the values observed for urinary 5-HIAA in different stages of the disease, observing the time spent from the clinical onset of the disease and the correlation between the ALvarado score and the histological image of the resected specimens.
• What specific improvements should the authors consider regarding the methodology? What further controls should be considered?
The number of studied cases is large enough to allow some statistical correlations, but as the authors already observed, the lack of a control group could be an issue in interpreting the usefulness of the results.
• Are the conclusions consistent with the evidence and arguments, and do they address the central question?
The study's conclusions are consistent with the evidence and arguments presented, but more studies are necessary regarding the clinical utility of measuring the spot values of urinary 5-HIAA.
• Any additional comments on the tables and figures?
No additional comments on tables and figures
Author Response
Dear Reviewer,
Thank you very much for your valuable evaluation and insightful comments.
In our study, we aimed to identify early changes that could yield novel and distinguishable results compared to previous research. As you correctly noted, a significant difference was observed within the first 12 hours.
To further investigate this early-phase difference, a new study including a control group has been designed and submitted for ethical approval.
Sincerely,
Dr Murat Demir
Reviewer 2 Report
Comments and Suggestions for Authors
Dear colleagues, thank you for your work! A very interesting scientific article, and I would really like to have reference values for determining threshold levels. However, I understand that this requires a significantly larger number of subjects. It is also interesting - did you include metallokinases in the comparison, for example?
Author Response
Dear Reviewer,
Thank you for your positive feedback on our study.
We would like to clarify that metallokinases were not included in the comparison in our research.
Best regards,
Dr Murat Demir
Reviewer 3 Report
Comments and Suggestions for Authors
please see the attached document.

Author Response
Dear Reviewer,
- Thank you for your positive feedback on our study. We observed an increase in 5-HIAA levels in the early phase. The rise in CRP levels also supports the progression of inflammation and infection.
- The early-phase difference we identified encouraged us to explore this further. As you pointed out, we have submitted a new study for ethical approval. This study includes a control group and aims to determine a diagnostic cut-off value for 5-HIAA.
- Previous studies have investigated changes in 5-HIAA levels in patients with acute appendicitis, with varying results. After reviewing these studies, we aimed to assess 5-HIAA levels in relation to histopathology and the Alvarado score. Our finding of a significant increase within the first 12 hours is novel and contributes to the literature. Following this result, we designed a new study to identify a reliable cut-off value through comparison with a control group, which has been submitted for ethical approval.
We sincerely appreciate your evaluation and constructive comments. However, we respectfully ask you to reconsider the “not adequate” assessment. This study demonstrates a promising finding—the early rise in 5-HIAA levels—which we believe lays the groundwork for further investigation. We are excited about these results and are currently initiating the next phase of our research.
Sincerely,
Dr Murat Demir
Round 2
Reviewer 3 Report
Comments and Suggestions for Authors
I accepted the authors' explanation that you have conducted a new investigation trying to answer the key questions in my previous review, i.e., the control group and the possible cut-off value of urinary 5 HIAA in suspected acute appendicitis.
For making consistency with this unmature results, I suggested that the topic of this manuscript could be adjusted to be a little more humble, such as 「Urinary 5-HIAA as a Potential Diagnostic Marker in Acute Appendicitis: A Preliminary Report of Promising Role in Early Detection」.
Author Response
Dear Editor,
We are grateful for your evaluation and contributions to the article. Based on your suggestions, the title of the article was revised to "Urinary 5-HIAA as a Potential Diagnostic Marker in Acute Appendicitis: A Preliminary Report of Promising Role in Early Detection". The changes are marked in the main text.
Best regards,
Dr. Murat Demir